# Isolated pairs of Majorana zero modes in a disordered superconducting lead monolayer

Gerbold C. Ménard[1,4], Andrej Mesaros [2], Christophe Brun[1], François Debontridder[1], Dimitri Roditchev[1,3], Pascal Simon [2] & Tristan Cren[1]

Majorana zero modes are fractional quantum excitations appearing in pairs, each pair being a building block for quantum computation. Some signatures of Majorana zero modes have been reported at endpoints of one-dimensional systems, which are however required to be extremely clean. An alternative are two-dimensional topological superconductors, such as the Pb/Co/Si(111) system shown recently to be immune to local disorder. Here, we use scanning tunneling spectroscopy to characterize a disordered superconducting monolayer of Pb coupled to underlying Co-Si magnetic islands. We show that pairs of zero modes are stabilized: one zero mode positioned in the middle of the magnetic domain and its partner extended all around the domain. The zero mode pair is remarkably robust, isolated within a hard superconducting gap. Our theoretical scenario supports the protected Majorana nature of this zero mode pair, highlighting the role of magnetic or spin-orbit coupling textures.

[1] Institut des NanoSciences de Paris, Sorbonne Université and CNRS-UMR 7588, 75005 Paris, France. [2] Laboratoire de Physique des Solides, CNRS, Univ. Paris-Sud, Université Paris-Saclay, 91405 Orsay Cedex, France. [3] Laboratoire de physique et d'étude des matériaux, ESPCI PSL and CNRS-UMR 8213, 75005 Paris, France. [4] Present address: SPEC, CEA, CNRS, Université Paris-Saclay, CEA Saclay, 91191 Gif-sur-Yvette Cedex, France. Correspondence and requests for materials should be addressed to P.S. (email: tristan.cren@upmc.fr) or to T.C. (email: pascal.simon@u-psud.fr)

As intrinsic topological superconducting materials seem rare in nature, the main guideline for finding Majorana zero modes (MZM) is to induce topological superconductivity by combining spin–orbit coupling, Zeeman field, and conventional superconductivity. Such recipe has been applied in one-dimensional (1D) systems consisting of semiconducting wires[1–3], chains of magnetic adatoms[4–7], and confined electron gas[8]. Evidences of MZM have been reported as zero-bias peaks in transport and tunneling experiments. However, finding unambiguous signatures of MZM remains challenging because of the presence of other unavoidable Yu-Shiba-Rusinov or Andreev-like bound states[9] whose removal demands fine-tuning and extremely clean systems. In two-dimensional systems, instead of MZM, 1D dispersive chiral Majorana fermions are theoretically expected at the sample edges[10]. Signatures of such 1D chiral Majorana mode have recently been reported in STM measurements around nanoscale magnetic islands either buried below a single atomic layer of Pb[11], adsorbed on a Re substrate[12], and also in transport experiments in two-dimensional (2D) heterostructures made of a quantum anomalous Hall insulator bar contacted by a superconductor in a magnetic field[13]. Interestingly, Majorana zero modes are also predicted in 2D as point-like excitations (Fig. 1a)

bound to superconducting vortices in superfluid $^3$He[14], in p-wave superconductors[15,16] and in fractional quantum Hall state[17]. Direct experimental observations of MZM in vortex cores in topological surface states coupled to a bulk superconductor were recently reported[18–20]. Generically the vortex core MZMs are swamped by Caroli-Matricon-de Gennes low-energy excitations with a typical energy level spacing $\Delta^2/E_F$[14,21], which can only be removed by tuning the Fermi energy $E_F$ close to the gap energy $\Delta$, yet there remains a substantial sensitivity to disorder in these reported works[18–20]. In 2D topological superconductors Majorana zero modes are not expected to reside only on point-like defects, such as vortices. It is in fact predicted that in systems containing an odd number of point-like MZMs, another extended MZM must appear at the system's edge because Majorana zero modes must appear in pairs (Fig. 1b). Thus, finite size topological superconductors are ideal systems to get MZM with a very large spatial extent, their wavefunction extending all around the topological domains boundary. In this work, we report the observation of a Majorana pair constituted of a MZM located on a point-like defect paired to a rim-like MZM. This MZM pair appears insensitive to strong crystalline disorder and isolated in energy by a hard superconducting gap.

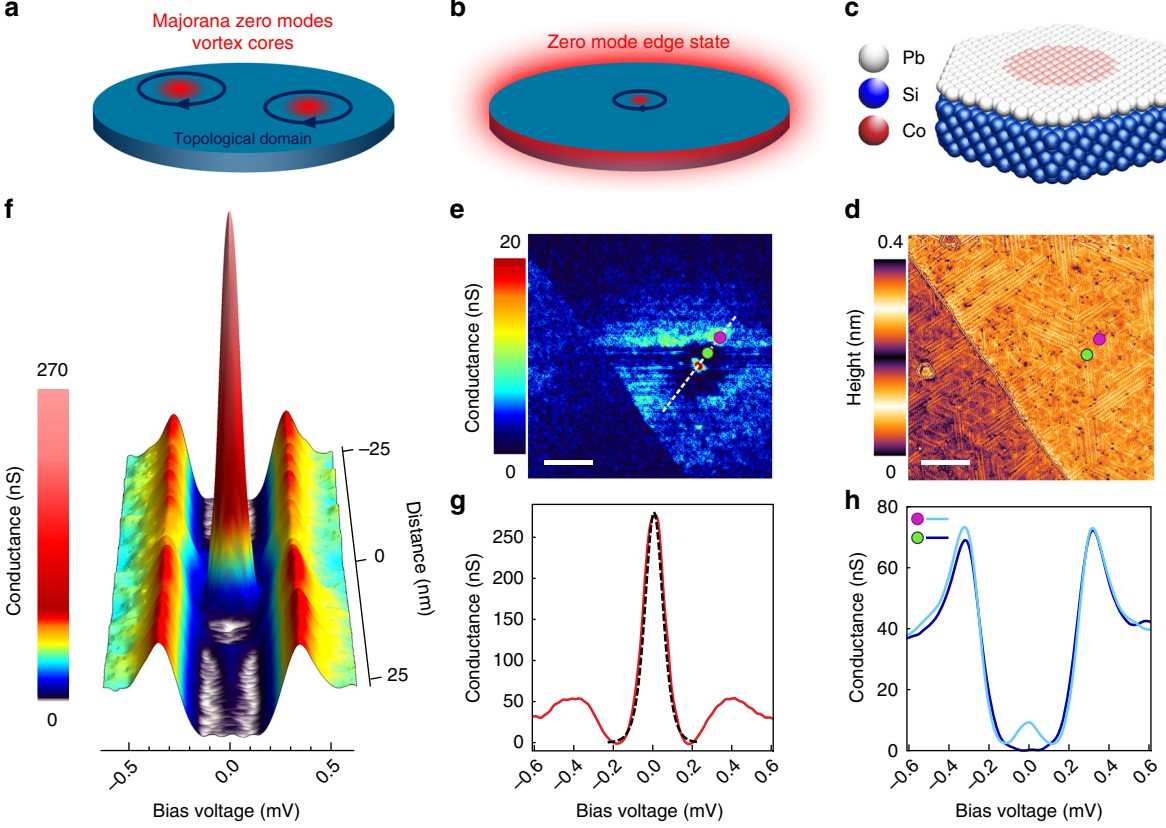

**Fig. 1** Spectroscopy of a pair of Majorana zero modes in atomic Pb monolayer. **a** Schematic view of a pair of vortices in a topological superconductor, each carrying a single Majorana zero mode. **b** Schematic view of a single vortex whose Majorana zero mode has to be paired with a Majorana zero mode surrounding the topological domain. **c** Schematic structure of the Pb/Co/Si(111) samples: a Co–Si magnetic domain is buried below a Pb monolayer. **d** Scanning tunneling microscopy image (scale bar is 20 nm) of the sample showing a Pb monolayer with devil's staircase structure ($V_T = 50$ mV, $I_T = 50$ pA). The underlying Co–Si island doesnot appear in the topography. **e** Conductance map (scale bar is 20 nm) at $V_T = 0$ mV showing a domain with a strong zero-bias peak dot (in red) surrounded by a gapped region (dark blue) itself surrounded by a zero-bias rim (light blue). **f** Conductance spectra along the linecut marked in **e** showing a very strong peak within the gap surrounded by a gapped area and another zero-bias peak (blue) outside of the domain. **g** Conductance curve taken at the center of the domain showing a very high zero-bias peak (Red curve). The dashed black curve is a fit with a state at $E \approx 6\,\mu$V with an electronic temperature of 350 mK. This can be considered as a zero-energy state within the precision of the experimental set-up. **h** The light blue conductance curve is taken on the domain rim (light blue in **e**) shows a peak at zero bias. The dark blue conductance curve is taken inside the dark blue region inside the domain, it shows a hard gap with no peak at zero bias. The green and purple markers in **e** and **d** indicate the position at which those spectrum were taken

## Results

**STM measurements**. Our system is based on a monolayer of Pb/Si(111) with a strong Rashba spin–orbit coupling that induces spin-triplet correlations, this material remaining however a trivial superconductor[22–24]. In our previous work we showed that applying a local magnetic exchange field with the help of a buried self-assembled Co–Si nano-island (see the structure in Fig. 1c) induces a topological transition evidenced by the presence of in-gap dispersive edge states around the island[11]. In contrast to this previous work, here, the islands are grown to larger sizes $D \sim 15–20$ nm (instead of $D \sim 5–10$ nm in previous work). We probe the system using scanning tunneling microscopy (STM) and spectroscopy (STS) with a normal Pt tip.

In Fig. 1d we show the topography of the surface over an island, finding no distinguished local features. Indeed, the magnetic islands are buried into the Si substrate under the Pb monolayer. However, the islands distribution can be revealed after annealing[11]. The presence of underlying magnetic clusters becomes apparent by acquiring spectroscopic maps at the Fermi energy, well inside the gap. In contrast with the featureless topography, the corresponding zero-bias conductance map in Fig. 1e displays strong spectroscopic features, which are absent in the reference system grown without islands. A very local feature of size ~1 nm, much smaller than the coherence length $\xi \approx 50$ nm, appears as a red spot in the center of the domain. Another feature follows the edge of the domain (light blue rim), decaying sharply towards the inside and decaying slowly over a few tens of nanometers outside of the domain. We attribute the inner diameter of this ring-like feature to the size $D$ of the underlying magnetic Co–Si island. This behavior is observed in at least five different locations with $D \sim 15–20$ nm (two others are displayed in the Supplementary Fig. 2). The radial linecut in Fig. 1f shows that the central feature is associated with a very strong peak located very close to zero-energy in a well developed energy gap. The edge feature, appearing as a blue halo in Fig.1e, is also associated with a zero-energy peak located in a hard energy gap. These two features are separated by a dark corona where a hard gap is restored without any measurable signal at zero bias (see spectrum in Fig. 1h). The energy width of the central peak is well captured by an electronic temperature of 350 mK, which is close to the 320 mK base temperature of our microscope (see Fig. 1g and Supplementary Fig. 1).

A typical spectrum taken at the edge, Fig. 1h, shows a similarly thermally broadened peak. Note that the peak height is higher for the central feature than for the edge one (Fig. 1g, h), which is expected for a pair of modes. Namely, if each feature represents a single normalized wavefunction, the density of states each contributes should be the same, which is a sum rule. One implication of this sum rule is that the local density of states must be lower for the edge feature because of its extension over a larger spatial area, just as we observe experimentally. The sum rule is quantitatively true in the theoretical models below, but the quantification in the experimental data is unfortunately fruitless due to limited energy resolution.

The magnetism of the Co–Si clusters we study was well established in refs. [25–27] and its effect on superconductivity was discussed in our previous work[11]. The effect of the Co–Si clusters is very different from the one of a collection of single Co impurities. Indeed, single magnetic atoms in the superconducting Pb monolayer gives rise to the formation of well identified Yu-Shiba-Rusinov bound states[28]. If in the present study the islands acted as bunches of non-ordered magnetic impurities, we would expect to observe many YSR states randomly distributed inside the superconducting gap, which we do not observe here.

Remarkably, in our system with islands we find only zero modes and no other states inside the gap. This therefore indicates a magnetic ordering on the islands due to an assembled Co–Si crystalline structure[25–27]. Furthermore, we find a locally well-defined gap everywhere (note that theoretically one expects that these zero modes are well protected from decoherence by the hard energy gap). The energetically isolated and spatially well-defined zero-bias peak pair in a disordered superconductor is strongly reminiscent of the Majorana zero-mode pair appearing when a single point-like Majorana is paired with another one extended along the boundary of a topological domain. Based on these facts we interpret that the domain seen in our spectroscopic map is induced by a magnetically ordered Co–Si island located below the Pb layer. To further support this interpretation, we consider theoretical scenarios, which explain all the key features: (1) appearance of a point-like zero mode, and its corresponding ring-shaped pair, (2) energetical isolation of the pair within an unperturbed energy gap, (3) decay length of wavefunctions differing on top and outside the island.

**Theoretical modeling**. The most natural theoretical explanation relies on a superconducting vortex located at the center of the island. This scenario proves entirely insufficient. Indeed, we directly observe superconducting vortices induced by external magnetic field, and their zero-bias conductance maps show the vortex core states extending over the largest spatial scale $\xi$, covering an area larger than entirely the Co–Si islands area (see Supplementary Fig. 3). This is in stark contrast to the observed zero-mode pair, while consistent with simple theoretical modeling (see Supplemementary Fig. 4). Furthermore, in a vortex the superconducting energy gap is filled with many Caroli-Matricon-de Gennes states[14,21], their number in the gap being expected to be $E_F/\Delta_S \sim 2000$ according to the values of Fermi energy $E_F \sim 660$ meV[29] and superconducting bulk gap $\Delta_S \approx 0.35$ meV. This is again in stark contrast to our zero-mode pair observations. Anyhow, excluding a superconducting vortex, the robust zero mode in the center of the island is still indicative of some underlying topological defect, motivating our next theoretical scenario.

We now consider a vortex-like topological defect in the phase of the spin–orbit coupling, rather than in the phase of the superconducting order parameter. This scenario offers strong agreement with our observations. A minimal model in the Bogoliubov-de Gennes formalism for quasiparticle excitations of the superconductor reads as:

$$\hat{H} = \int d^2\mathbf{r}\, \Psi_\mathbf{r}^\dagger \big[ (-\eta\nabla^2 - \mu)\tau_z + V_z(\mathbf{r})\sigma_z + \Delta_S\tau_x \big]\Psi_\mathbf{r} \\ + \mathcal{H}_{SO-defect}(\mathbf{r}), \tag{1}$$

written in the standard basis $\Psi_\mathbf{r} \equiv \big( c_{\mathbf{r}\uparrow}, c_{\mathbf{r}\downarrow}, c_{\mathbf{r}\downarrow}^\dagger, -c_{\mathbf{r}\uparrow}^\dagger \big)^T$, where $c_{\mathbf{k}a}$ annihilates an electron of momentum $\mathbf{k} = (k_x, k_y)$ and spin $z$-component $a = \uparrow, \downarrow$, with Pauli matrices $\sigma_a$, $a = x, y, z$, acting in spin-space and $\tau_a$, Pauli matrices mixing electrons with holes. The term for a vortex in the spin–orbit coupling is:

$$\mathcal{H}_{SO-defect} = c_{\mathbf{r}\uparrow}^\dagger \big\{ \alpha e^{i\theta(\mathbf{r})}, \nabla_x - i\nabla_y \big\} c_{\mathbf{r}\downarrow} + h.c., \tag{2}$$

with $\theta(\mathbf{r})$ the polar angle and $\{,\}$ denotes the anticommutator. Without the $exp(i\theta(\mathbf{r}))$ phase factor (see Supplementary Note 4), Eq. (2) is just the standard Rashba spin–orbit coupling term of the Pb monolayer on the Si substrate, which in our system is $\alpha \sim 300$ meV. Here, we assume that the spin–orbit magnitude remains constant, while the vortex-like winding of the phase changes the local angle of the momentum-spin locking. The model of Eqs. (1) and (2) with a constant magnetic exchange field

$V_z(\mathbf{r}) = V_z$ has been introduced with the exclusive focus on the existence of a point-like Majorana zero mode[30,31], which is found in the case of topologically non-trivial bulk superconductivity $V_Z^2 > \Delta_S^2 + \mu^2$. In contrast, in this work we find both the full excitation spectrum and the zero modes by using several system geometries and numerical models (see Methods). We model the Co–Si island of size $D$ by setting the magnetic exchange $V_z(\mathbf{r})$ non-zero only within a disk of radius $R = D/2$ and zero outside. We find a striking resemblance with our experimental observations (see Fig. 2 and Methods): (1) The zero-energy density of states shows a very narrow peak at the island's center (position of the spin–orbit vortex), as well as a ring-like shape extending outside the island at larger lengthscale (Fig. 2a); these are exactly two Majorana wavefunctions. The wavefunctions decay inside the island at a short lengthscale, which depends on a magnetic length $l_V \sim 1/V_z$, while outside the island they decay on the much larger lengthscale $\xi$ (see Fig. 2b and Supplementary Fig. 6). (2) The thermally broadened local density of states at the island center, at the island edge, and far away (Fig. 2c, d) show preserved coherence peaks. In the center and at the island edge, a zero-energy peak is present while it is absent away from these regions. All this matches perfectly our observations in Fig. 2g, h. (3) The Majorana zero-mode pair appears within a fully preserved superconducting energy gap only under the condition of a strong enough $V_z$ (see Fig. 2e). Note that the topological gap itself is expected to be large (near $\Delta_S$) in the absence of defects even for high $V_z/\Delta_S$, a consequence of the large spin–orbit coupling $\alpha > \Delta_S$ independent of dimensionality. The Majorana zero-mode pair remains robust for a range of island sizes (see Methods). The hard gap is generally expected to protect the pair against various spatial disorder, yet we believe this deserves a separate theoretical study.

Although the reason for the appearance of a spin–orbit defect is not obvious, we note that (1) The disordered Pb layer breaks all spatial symmetries, which allows the mixing of different momentum-spin locking angles by the defect (see Supplementary Note 5), and (2) Inhomogeneous spin–orbit textures are naturally linked to inhomogeneous magnetic exchange fields[32–34]. Theoretically, it is possible to replace the spin–orbit defect term in the model, Eq. (2), by a locally varying exchange field texture:

$$\hat{H}_{\text{texture}} = \int d^2\mathbf{r} \Psi_{\mathbf{r}}^{\dagger} [\mathbf{V}(\mathbf{r}) \cdot \sigma] \Psi_{\mathbf{r}}. \qquad (3)$$

In fact, by rotating the spin-axis of the electrons, this model can be rewritten in a form similar to the model in Eq. (1) with a homogeneous spin–orbit coupling term but with a magnetic texture (see Supplementary Note 6). Such a mapping between inhomogeneous magnetism and spin–orbit coupling is well-known for one-dimensional systems[32]. In 2D, it was shown that skyrmion magnetic textures $\mathbf{V}(\mathbf{r})$ in Eq. (3) can host a Majorana zero mode at their center[35,36], but in that case the superconducting gap is strongly reduced by the presence of low-energy excitations. In the Supplementary Note we detail a model based on a skyrmion texture in Eq. (3), showing that this model reproduces the key observed features as successfully as the spin–orbit vortex model. Thereby we give an example of a previously unexplored connection between two theoretical concepts in 2D: a topological defect in spin–orbit coupling and an inhomogeneous magnetic exchange texture. We demonstrate that both of these theoretical concepts give rise to remarkable features, which match our experimental observations, namely: (1) A pair of Majorana zero modes, one point-like and one ring-shaped, (2) Zero modes isolated in a hard superconducting gap, and (3) Zero-mode wavefunctions localized over two very different lengthscales.

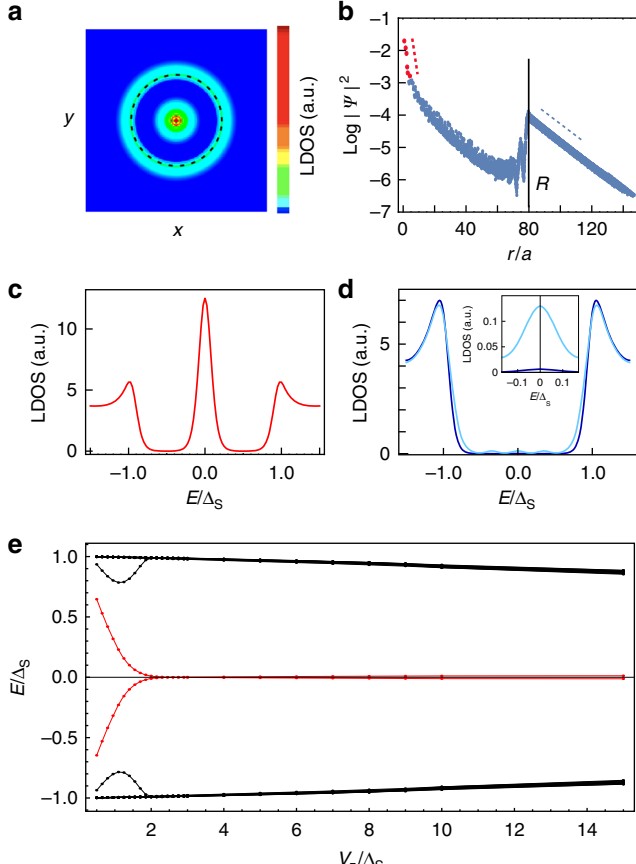

**Fig. 2** Theory of spin–orbit defect and magnetic exchange. **a** Map of the local density of states (LDOS) at zero energy in the model of Eq. (1) solved in a circular system with a single spin–orbit vortex at the center of a disk (dashed line), which has constant magnetic exchange $V_z = 5\Delta_S$, and $V_z = 0$ outside, while all system parameters are $(\xi/a, R/a, l_V/a, l_F/a, l_{SO}/a) = (400; 300; 80; 45; 3.3)$ (see Methods for definitions and technical details), where superconducting coherence length $\xi$ is larger than disk radius $R$, and system size is $L = 8000$. This LDOS is contributed by exactly two Majorana zero-mode wavefunctions. **b** Angularly averaged amplitude of the wavefunction at zero energy (dashed lines are guide to eye). The superconducting coherence length $\xi/a = 20$ is smaller than disk radius $R/a = 80$ (vertical line) in a system of size $L = 300$. See Methods for definitions, parameter values and technical details of the two-dimensional tight-binding calculation of model Eq. (1). **c** Energy-dependent LDOS at the center of the island with a thermal broadening of $k_B T/\Delta_S = 0.1$. **d** Energy-dependent LDOS at the edge of the magnetic disk (light blue curve) and inside the magnetic disk at distance $0.75R$ from the center (dark blue curve). The inset show a zoom on the zero-bias density of states, revealing a well-defined zero-bias peak. **e** Spectrum of excitations in superconductor of pairing energy $\Delta_S$ for spin–orbit vortex—anti-vortex pair in a plane of size $L = 450$ with periodic boundary conditions and with constant magnetic exchange $V_z$ (see Methods). The lowest energies are plotted for each $V_z$, with lowest two in red. Above a critical magnetic exchange two zero-energy states are localized at the two spin–orbit vortices

## Discussion

Our work demonstrates the first example of a zero-mode pair that is robust to spatial disorder, as well as protected by a hard superconducting energy gap. Strikingly, one of the modes has a ring-shaped wavefunction fully reliant on the two-dimensionality of the system. Based on our theoretical modeling, which matches the key features of the data, we interpret the zero-mode pair as a Majorana pair. This opens completely new possibilities for

spatially accessing and manipulating quantum information with Majorana pairs. The possible novel consequences of extended wavefunctions on the braiding and fusion of Majorana states should motivate in-depth theoretical studies. Since the spin–orbit vortex model is closely related to a magnetic texture model, we expect that a magnetic characterization of the islands is needed for a deeper understanding of the system. This is challenging due to the buried nature of the Co–Si islands. A key experimental advantage of our superconductor layer is that it is deposited on a Si substrate, which avoids the well-known problems of contacting to an underlying metallic substrate or bulk superconductor. On the other hand, there is no obvious tuning parameter for a topological transition, and real-time manipulation of the system seems challenging. Nevertheless, strong local electric fields are expected to be useful, while the Co islands could be patterned and the Si substrate is amenable to circuitry design. More generally, our results bring into focus the necessity of understanding and developing structures that combine inhomogeneous magnetism with superconductivity as a promising platform for quantum computation.

## Methods

**Sample preparation**. The $7 \times 7$ reconstructed n-Si(111) (room temperature resistivity of few m$\Omega$ cm) was prepared by direct current heating to 1200 °C followed by an annealing procedure driving the temperature from 900 to 500 °C. Subsequently, $1.1 \times 10^{-2}$ monolayer of Co were evaporated in 10 s on the $7 \times 7$ reconstructed Si(111) substrate kept at room temperature. The Co was evaporated from an electron beam evaporator calibrated with a quartz micro-balance. Four monolayers of Pb were then evaporated using another electron beam evaporator. The Pb overlayer was formed by annealing the sample at 375 °C for 4 min 30 s by direct current heating. This step leads to a combination of devil's staircase phases reconstruction of the Pb monolayer. At no stage of the sample preparation did the pressure exceed $P = 3 \times 10^{-10}$ mbar.

**Measurements**. The scanning tunneling spectroscopy measurements were performed in-situ in a home-made apparatus at a base temperature of 320 mK and in ultrahigh vacuum in the low $10^{-11}$ mbar range. Mechanically sharpened Pt/Ir tips were used. The bias voltage was applied to the sample with respect to the tip. Typical set-point parameters for topography are 20 pA at $V = -50$ mV. Typical set-point parameters for spectroscopy are 120 pA at $V = -5$ mV. The electron temperature was estimated to be 350 mK. The tunneling conductance curves $dI/dV$ were numerically differentiated from raw $I(V)$ experimental data. Each conductance map is extracted from a set of data consisting of spectroscopic $I(V)$ curves measured at each point of a $220 \times 220$ grid, acquired simultaneously with the topographic image. Each $I(V)$ curve contains 700 energy points in the $[-0.7; +0.7]$ meV energy range.

**Numerical modeling and calculations**. We adapt the model of Eq. (1) in three different ways to facilitate a numerical analysis. The different system geometries considered in the numerical solution of the spin–orbit vortex Eq. (1) are sketched in Supplementary Fig. 5.

As first approach, we consider a constant magnetic exchange disk at the origin, $V_z(|\mathbf{r}| < R) \equiv V_z$, and a spin–orbit vortex located at the origin. We use rotational symmetry in the plane to study a large system and access a wide range of lengthscales. The symmetry together with a standard rescaling of the Bogoliubov-de Gennes operators $\Psi_{\mathbf{r}} \equiv \left( c_{\mathbf{r}\uparrow}, c_{\mathbf{r}\downarrow}, c^{\dagger}_{\mathbf{r}\downarrow}, -c^{\dagger}_{\mathbf{r}\uparrow} \right)^T \equiv \frac{1}{\sqrt{r}} \exp(im\theta) \Psi_r$ reduces the Hamiltonian of Eq. (1) to a radial problem, which we straightforwardly discretize by $r = ja$, $j = 1, 2, ..., L$ on a chain of spacing $a$, into a tight-binding model:

$$\mathcal{H}^{TB}_{radial} = \sum_{j=1...L-1} \Psi^{\dagger}_{j+1} \left[ -t - \frac{i\alpha}{2a}\sigma_y \right] \tau_z \Psi_j + \sum_{j=1...L} \Psi^{\dagger}_j \left\{ \left[ -\mu + \frac{4m^2-1}{4j^2} + \frac{m\alpha}{j}\sigma_x \right] \tau_z \right.$$
$$\left. + \Delta_S \tau_x + V_z(j)\sigma_z \right\} \Psi_j,$$
(4)

where the integer angular momentum is $m$. The shortcoming of the discretization is the treatment of the (polar and vortex) singularity at the origin: we choose the first site of the chain at $r = a$, while moving this position relative to $r = 0$ can strongly change the energetic contribution of the first sites of the chain. However, we find that the behavior at low-energy (up to superconducting pairing energy $\Delta_S$) is qualitatively robust. We perform exact diagonalization of Eq. (4), for system sizes (radii) up to 10.000 sites, combining angular momenta $m = -100, ..., 100$ since for typical parameters the lowest energy states at momenta $|m| > 50$ are consistently above energy $\Delta_S$.

The local density of states of Fig. 2a, c, d is presented in the lengthscale regime corresponding to experiments, namely for system radius $L$, superconducting coherence length $\xi$, island radius $R$, Fermi length $l_F$ and spin–orbit length $l_{SO}$:

$$L/\xi > 10$$

$$\xi/R > 1$$

$$R/l_F \sim 10$$

$$l_{SO} \lesssim l_F,$$

while the magnetic exchange energy $V_z$ was not accessible in these experiments. We estimate the energy-based lengthscales in tight-binding models as

$$\left( \frac{\xi}{a}, \frac{l_V}{a}, \frac{l_F}{a}, \frac{l_{SO}}{a} \right) = \left( \frac{t}{\Delta_S}, \frac{t}{V_z}, \frac{t}{E_F}, \frac{t}{\alpha} \right),$$
(5)

where $t$ is the nearest-neighbor hopping, and in Fig. 2a, c, d we choose precisely

$$(L, \xi/a, R/a, l_V/a, l_F/a, l_{SO}/a) = (8000; 400; 300; 80; 45; 3.3).$$
(6)

Note that the Fermi energy $E_F(\mu)$ is measured from the bottom of the band of the bulk clean model without the vortex, i.e., with spatially uniform spin–orbit coupling of amplitude $\alpha$. We put $\mu = 0$ so that the Fermi level is in the middle of the gap that opens at zero momentum in the clean bulk system.

The main result of this model with given island radius $R$ are two particle-hole partner wavefunctions $\psi_+(R)$, $\psi_-(R)$ at angular momentum zero, $m = 0$. They both have peaks at the center of the spin–orbit vortex and at the edge of island. Their energy splitting $E(R)$ quickly decays towards zero with growing $R$, and therefore in the limit of infinite island radius these wavefunctions would form an exact Majorana pair. Assuming that the exact Majorana zero modes $M_1$, $M_2$ are hybridized due to finite island size by the general Hamiltonian $\delta H(R) = iE(R)M_1 M_2$, one may assess the exact $M_1$, $M_2$ wavefunctions by inspecting $M_1 \equiv \psi_+(R) + \psi_-(R)$ and $M_2 \equiv i(\psi_+(R) - \psi_-(R))$. We find that the numerical wavefunctions of $M_1$ and $M_2$ are spatially separated and exactly localized at the center and at the island edge, respectively. This supports the claim of exact and truly separated Majorana pair created by the spin–orbit vortex and the island edge, so we freely use the term throughout. Note that for the quoted parameters, the energy splitting is $<10^{-2}\Delta_S$, much below our experimental energy resolution.

The Majorana pair is robust in a range of island sizes for the quoted parameters: (1) The mixing of zero modes due to their real space overlap leads to a noticeable energy splitting of size $>5\%$ of the gap only when $R$ decreases below $\xi/2$, (2) For increasing $R$ (also beyond $\xi$), although the zero-modes are stable, the dispersive edge modes move into the gap (see Supplementary Fig. 5). The approximately constant energy spacing of dispersive edge modes reduces inversely proportionate to the island radius, as quantization of waves on the edge directly predicts. For the above parameters, $R$ has to reach $2\xi$ before $>5$ dispersive edge states move into the gap. We also check the independence of our results on system size by varying the system radius $L$ so that the island size $R$ is between 5 and 70% of $L$.

The qualitative variation of the Majorana zero-mode wavefunctions with $V_z$ (see Supplementary Fig. 5) shows that both the central and the edge wavefunctions are strongly controlled by $l_V$ on the inside of island, but the $\xi$ lengthscale is also important, since $\xi$ dominates the edge zero-mode decay outside the island. The obvious Friedel oscillations at scale $l_F$ are not observed in the experiments due to the diffusive regime.

As a second approach, we study the straightforward two-dimensional tight-binding version of the continuum model in Eq. (1):

$$\mathcal{H}^{TB}_{2D} = \sum_{\mathbf{I}} \sum_{\delta = a\hat{x}, a\hat{y}} \Psi^{\dagger}_{\mathbf{I}+\delta} \left[ -t + \frac{i\alpha}{2a}\hat{\lambda}_\delta \right] \tau_z \Psi_{\mathbf{I}} + \sum_{\mathbf{I}} \Psi^{\dagger}_{\mathbf{I}} \left[ -\mu\tau_z + \Delta_S \tau_x + V_z(\mathbf{I})\sigma_z \right] \Psi_{\mathbf{I}},$$
(7)

where the Bogoliubov-de Gennes operators $\Psi_{\mathbf{I}} \equiv \left( c_{\mathbf{I}\uparrow}, c_{\mathbf{I}\downarrow}, c^{\dagger}_{\mathbf{I}\downarrow}, -c^{\dagger}_{\mathbf{I}\uparrow} \right)^T$ are defined on a two-dimensional square lattice $\mathbf{I} = (i, j)a$, $i = 1, ..., L$, $j = 1, ..., L$, of spacing $a$, and the $2\pi$-winding spin–orbit vortex term is given by

$$\begin{pmatrix} \hat{\lambda}_x \\ \hat{\lambda}_y \end{pmatrix} = \begin{pmatrix} -\sin(\theta_+) & -\cos(\theta_+) \\ \cos(\theta_+) & -\sin(\theta_+) \end{pmatrix} \begin{pmatrix} \sigma_x \\ \sigma_y \end{pmatrix},$$
(8)

where the counter-clockwise polar angle $\theta_+$ is defined from the center of a plaquette defining the position $\mathbf{P}_+$ of the $2\pi$ spin–orbit vortex. The system shape (Supplementary Fig. 5) is a torus, i.e., there are periodic boundary conditions, with $L$ up to 450. The geometry ensures there are no spurious edge states of the system itself and no issues with regularizing the vortex singularity, but necessitates, including a $-2\pi$ spin–orbit vortex (the corresponding polar angle $\theta_-$ runs clockwise), which we set at distance $L/2$ from the $2\pi$ one. We include the island as a single disk of non-zero $V_z(|\mathbf{I} - \mathbf{P}_+| < R) \equiv V_z$, with radius $R$ small enough so that the $-2\pi$ spin–orbit vortex is outside it where $V_z(\mathbf{I}) = 0$.

This approach corroborates that for $V_z$ larger enough than $\Delta_S$ there are two wavefunctions at energy zero, localized at the $2\pi$ spin–orbit vortex at the center of the island and at the island edge, while there are no features on the $-2\pi$-winding spin–orbit vortex (Fig. 2b). The spin–orbit vortex Majorana zero mode at the disk center has a peak localized at a much smaller lengthscale than its tail or the Majorana zero mode at the edge of the island, although quantification is impossible due to limited system sizes. The excited states below the pairing energy $\Delta_S$ are island edge states, which are sparse (see Supplementary Fig. 6). We checked that the numerical Majorana zero-mode wavefunctions are robust to dilute local random potential impurities, as expected from their topological protection. We vary the island radius from 10 lattice sites to 70% of the vortex—anti-vortex distance, reaching 100 sites. The shortcoming of this model is the small system size $L$, forcing the lengthscale $\xi$ to remain smaller than the island radius $R$. For Fig. 2b we set parameters

$$(L, \xi/a, R/a, l_V/a, l_F/a, l_{SO}/a) = (300; 20; 80; 2; 1.9; 0.7). \tag{9}$$

As a third approach, to isolate the properties of the central Majorana zero mode (Fig. 2e), we consider the model $\mathcal{H}_{2D}^{TB}$ where the magnetic exchange is constant in the entire plane, $V_z(\mathbf{I}) = V_z$, so that there are no edges whatsoever (see Supplementary Fig. 5). The Majorana zero modes at the $2\pi$ spin–orbit vortex and $-2\pi$ spin–orbit vortex centers are expected to appear as a pair only in the topological superconductivity regime $V_z > V_c = \Delta_S$ (since we take $\mu = 0$), which agrees with the numerical solution (shown in Fig. 2e) up to an expected shift in the critical value of $V_c$. Remarkably, there are no other states at energies below the bulk superconducting gap $\Delta_S$. We note that the bulk gap does close at the topological transition at $V_z = V_c$ in the thermodynamic limit, as evidenced numerically by the fact that bulk states near $V_c$ move down in energy as system size $L$ is increased. The exact parameters chosen here are:

$$(L, \xi/a, l_V/a, l_F/a, l_{SO}/a) = (450; 80; 8; 1.9; 0.7). \tag{10}$$

## Data availability

The datasets generated and analyzed during the current study are available from the corresponding author (T. C.) upon reasonable request.

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

## Acknowledgements

This work was supported by the French Agence Nationale de la Recherche through the contract ANR Mistral, and by the Région Ile de France through the DIM Nano-K project ETERNAL. G.C.M. acknowledges funding from the CFM foundation.

## Author contributions

G.C.M., C.B., and T.C. carried out the experiments. G.C.M. and T.C. processed and analyzed the data. A.M. and P.S. performed the theoretical study. A.M., P.S., and T.C. wrote the manuscript with contributions from the others. D.R., T.C., and F.D. designed and built the STM set-up.

## Additional information

**Competing interests:** The authors declare no competing interests.

