## [Peer Review File · Nature Communications]

Reviewers' comments:

Reviewer #1 (Remarks to the Author):

The revised version is much improved. I recommend it publish in Nature Communications.

Reviewer #3 (Remarks to the Author):

In this resubmission the authors address most of the issues raised in the previous round. However, the explanation regarding the shift of the zero bias peak away from zero energy is confusing and, possibly, wrong. More specifically, the authors state that “... *the splitting enforces states at $\pm E$ and one typically obtains the results as we do. This type of finite-size splitting of Majorana zero modes is common in the theoretical study of 1D and 2D models.*” The statement seems to suggest that finite size splitting results in one Majorana mode having energy $-E$, while the other one has energy $+E$. In fact, this is definitely not the case. If two (localized) Majorana modes (described by the wave functions ϕ_A and ϕ_B) hybridize due to finite spatial separation, one obtains two (fermionic) Bogoliubov modes: $\psi_{+E} = \phi_A + i\phi_B$ and $\psi_{-E} = \phi_A - i\phi_B$ (up to normalization factors). Note that the fermionic modes at $+E$ and $-E$ are not localized. Assume, for example, that ϕ_A is the central Majorana (with small amplitude on the boundary) and ϕ_B is the boundary Majorana (with small amplitude in the center). The ψ_{+E} and ψ_{-E} modes have equal amplitudes both in the center, as well as on the boundary. Consequently, probing the center (or the boundary) should generate a symmetric zero-bias peak (or a split peak, with enough resolution), not a shifted peak. Of course, it is possible that the fermionic modes ψ_{+E} and ψ_{-E} be localized themselves, in the center and on the boundary, respectively, but in this case the corresponding Majorana modes $\phi_A = \psi_{+E} + \psi_{-E}$ and $\phi_B = i(\psi_{+E} - \psi_{-E})$ are delocalized and, practically, on top of each other (hence they are not “true”, well-separated Majorana bound states). Consequently, if the energy shifts of the zero-bias peaks ($+E$ in the center and $-E$ on the boundary) are real effects associated with the energies of the corresponding (localized) modes, this is a clear signature of those states not being Majorana zero modes, but rather trivial Andreev bound states. Given that this issue affects the key finding of this work, I believe that it should be clarified before the manuscript is published. If this is not possible, I still believe that the work is interesting enough to deserve consideration, but the claims (and the title) should be substantially revised.

Reply to Reviewer #3

In this resubmission the authors address most of the issues raised in the previous round. However, the explanation regarding the shift of the zero bias peak away from zero energy is confusing and, possibly, wrong. More specifically, the authors state that “... *the splitting enforces states at $\pm E$ and one typically obtains the results as we do. This type of finite-size splitting of Majorana zero modes is common in the theoretical study of 1D and 2D models.*” The statement seems to suggest that finite size splitting results in one Majorana mode having energy $-E$, while the other one has energy $+E$. In fact, this is definitely not the case.

➤ We apologize that our statement was too shortened and gave the wrong impression. In fact, we fully agree with the Reviewer's observations below and thank him/her for pointing out the misleading claims.

➤ Our theoretical models give splittings that correspond exactly to the Reviewer's expectations for Majorana zero modes. Below we explicitly state our findings and show that they agree both with Reviewer's and with general expectations for splitting of Majorana's in our formalism.

➤ Our experimental resolution is not high enough to undoubtedly reveal the splittings. Within resolution the experimental peak is consistent with a small splitting expected from theory, but this is not a quantitatively unique fit, nor can it be due to resolution. For completeness of presentation, we perform a fit with a split peak (motivated by theory) as well as a fit with one peak at zero energy, as a guide for future experiments concerned with values of splitting energy.

➤ We have edited the Methods section "Numerical modeling and calculations" and the Supplementary section "Measurements" (text in red) to include our complete and precise claims, which are presented below and in full accord with Reviewer's expectations.

If two (localized) Majorana modes (described by the wave functions f_A and f_B) hybridize due to finite spatial separation, one obtains two (fermionic) Bogoliubov modes: $\gamma_{+E}=f_A+if_B$ and $\gamma_{-E}=f_A-if_B$ (up to normalization factors). Note that the fermionic modes at $+E$ and $-E$ are not localized. Assume, for example, that f_A is the central Majorana (with small amplitude on the boundary) and f_B is the boundary Majorana (with small amplitude in the center). The γ_{+E} and γ_{-E} modes have equal amplitudes both in the center, as well as on the boundary. Consequently, probing the center (or the boundary) should generate a symmetric zero-bias peak (or a split peak, with enough resolution), not a shifted peak.

➤ Indeed this corresponds exactly to our theoretical findings. Below the superconducting gap, in our Bogoliubov-de Gennes formalism we find exactly two Bogoliubons, γ_{+E} and γ_{-E} , at energies $+E$ and $-E$, respectively. They are in the zero angular momentum channel and checked to be particle-hole partners, therefore in the limit $E \rightarrow 0$ they would form a Majorana pair. The main question then concerns the localization in real space.

➤ The figure exemplifies our results with typical parameters. The two Bogoliubons γ_{+E} and γ_{-E} have the same wavefunction amplitude (i.e., contribution to LDOS), which has two peaks, one at origin and one at edge of island (site 0 and site 300). This exactly matches the Reviewer's expectation. To consider the original Majorana wavefunctions which were hybridized due to finite size of island, we invert the mapping written by the Reviewer above (he/she writes the inversion below): $M_1=1/2(\gamma_{+E} + \gamma_{-E})$ and $M_2=i/2(\gamma_{+E} - \gamma_{-E})$. The figure shows the wavefunction amplitudes for M_1 and M_2 : they are spatially separated, clearly localized at origin and at island edge, respectively. This is strong evidence that our system has a true, spatially separated Majorana pair, which is hybridized by finite system size. To further confirm this, we have checked that the splitting E goes towards zero very quickly with larger island radius R . Conversely, the spatial separation of M_1 and M_2 is still clear for a twice smaller island ($R=150$) although the splitting grows to $E=6\%$ of the gap Δ_S .

➤ We note that on very general grounds, finite size effects mix an energetically isolated, spatially separated Majorana pair m_1, m_2 through the generic Hamiltonian $\delta H = i * E * m_1 * m_2$, which gives a hybridization precisely in the form discussed here.

➤ Please note that the theoretical splitting found for relevant parameters (e.g., figure) is below 0.1% of the pairing gap, far below our experimental resolution. Therefore, it is not surprising that we cannot reliably extract from the experimental LDOS the expected peaks (i.e., +E/-E double peak at both the origin and edge of island).

Of course, it is possible that the fermionic modes y_{+E} and y_{-E} be localized themselves, in the center and on the boundary, respectively, but in this case the corresponding Majorana modes $fA = y_{+E} + y_{-E}$ and $fB = i(y_{+E} - y_{-E})$ are delocalized and, practically, on top of each other (hence they are not “true”, well-separated Majorana bound states). Consequently, if the energy shifts of the zero-bias peaks (+E in the center and -E on the boundary) are real effects associated with the energies of the corresponding (localized) modes, this is a clear signature of those states not being Majorana zero modes, but rather trivial Andreev bound states. Given that this issue affects the key finding of this work, I believe that it should be clarified before the manuscript is published. If this is not possible, I still believe that the work is interesting enough to deserve consideration, but the claims (and the title) should be substantially revised.

➤ Clearly this scenario is completely wrong for our theoretical results, but was suggested by our imprecise writing in the previous version. Our expanded and more precise rewriting of the discussion of splitting in the Methods and Supplementary is hopefully satisfactory for the Reviewer and the issue is completely resolved.